# FROM SPARSE TO DENSE: LEARNING TO CONSTRUCT 3D HUMAN MESHES FROM WIFI

## ABSTRACT

Estimating the pose and shape of multiple individuals in a scene is a challenging problem. While significant progress has been made using sensors like RGB cameras and radars, recent research has shown the potential of WiFi signals for pose estimation tasks. WiFi signals offer advantages such as resilience to obstructions, lighting independence and cost-effectiveness. This raises the question of whether the sparse Channel State Information (CSI) of WiFi signals, with its limited size, can be utilized to regress dense multi-person meshes. In this paper, we introduce WiMTR (WiFi-based Mesh Regression Transformer), a novel end-to-end model for multi-person mesh regression using WiFi signals. WiMTR comprises four key components: CSI feature extractor, CSI feature encoder, coarse decoder and refine decoder. The CSI feature extractor captures channel-wise features, while the CSI feature encoder extracts global features through internal interactions. The coarse decoder regresses initial parameters using randomly initialized queries and the refine decoder further enhances the pose and shape parameters through a differentiation-based query generation approach. To facilitate our research, we curate a dataset specifically for multi-person mesh regression from CSI signals. The dataset consists of 171,183 frames, encompassing diverse scenes and multi-person scenarios. WiMTR achieves competitive results, with a Mean Per Joint Position Error (MPJPE) of 71.4mm, Procrustes Aligned MPJPE (PAMPJPE) of 29.7mm and Procrustes Aligned Vertex Error (PVE) of 57.3mm. WiMTR represents the first WiFi-based multi-person mesh regression framework and we plan to release the code and dataset to facilitate further research in this area.

## 1 INTRODUCTION

Estimating the pose and shape of multiple individuals in a scene is a highly active research area in machine learning. Significant progress has been made in this field, particularly in the realm of computer vision, where researchers have focused on estimating 3D human pose and shape from monocular images. However, monocular images inherently lack depth information, which is crucial for accurate 3D tasks. Additionally, image-based methods are sensitive to factors like occlusion and lighting conditions, making the pose estimation problem ill-posed.

In recent years, WiFi signals have emerged as an interesting alternative for pose estimation. WiFi-based sensors offer advantages such as immunity to occlusion and lighting, along with the added benefit of low deployment costs. Researchers have developed several WiFi-based sensors, including Wi2Vi (Kefayati et al., 2020), Person-in-WiFi (Wang et al., 2019b) and WiPose (Jiang et al., 2020), which have shown promising results in 2D and 3D pose estimation tasks. However, existing WiFi-based methods primarily focus on estimating 2D and 3D poses and neglect the simultaneous regression of pose and shape for human meshes. This raises the question of whether the sparse Channel State Information (CSI) of WiFi signals, which typically consists of only about 10,000 elements if flattened into a vector, can be leveraged to regress dense meshes (with a size of $N \times 6890 \times 3$) representing the pose and shape of multiple individuals. The key challenge lies in determining whether the CSI signals contain sufficient information to capture both pose and shape details of the human body.

In this paper, we address this research gap by introducing WiMTR (WiFi-based Mesh Regression Transformer), a novel fully end-to-end model for regressing the pose and shape parameters of multi-person meshes using WiFi signals. WiMTR comprises four main components: the CSI feature

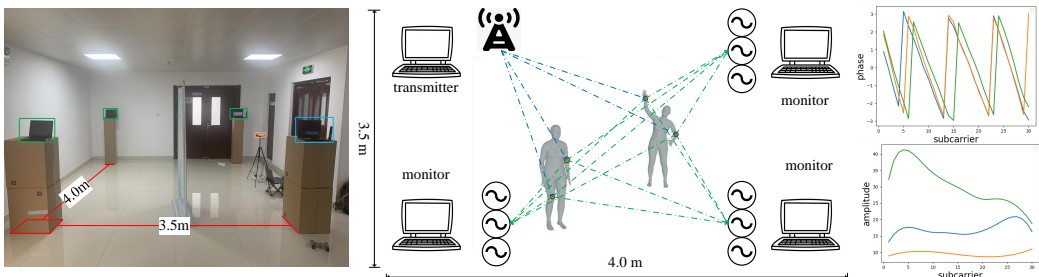

Figure 1: **Data Collection.** a) The left figure depicts a realistic scene used for data collection. The transmitter, located in the blue box, is a laptop, while the monitors are located in the green boxes. The Microsoft Azure Kinect camera is positioned in the yellow box. b) The middle figure depicts a schematic diagram of our data collection process. Each monitor is equipped with three antennas to receive signals transmitted by the laptop acting as the transmitter. The signals reflected by the human body are captured and used for human mesh regression. c) The right figures are schematic diagrams of the amplitude and phase waveforms of the received signals.

extractor, CSI feature encoder, coarse decoder and refine decoder. The CSI feature extractor captures channel-wise features from the amplitude and phase of the CSI signals and combines them into feature tokens. The CSI feature encoder then encodes the CSI token sequence and extracts global information. The coarse decoder regresses initial pose and shape parameters from the global features, while the refine decoder further refines these parameters using an identity token and joint queries. To facilitate our research, we have collected a dataset specifically for multi-person mesh regression using WiFi signals. The dataset consists of a substantial number of frames, providing rich data on multi-person scenarios and diverse scenes. Our experiments demonstrate that WiMTR achieves competitive results, with centimeter-level errors, including the Mean Per Joint Position Error (MPJPE) of 71.4mm, the Procrustes Aligned MPJPE (PAMPJPE) of 29.7mm and the Procrustes Aligned Vertex Error (PVE) of 57.3mm.

The main contributions of this paper can be summarized as follows:

- We collect a mesh dataset based on WiFi signals. This dataset includes a training set of size 151,838 and a test set of size 19,345.

- We present an fully end-to-end WiFi-based Mesh Regression Transformer, termed WiMTR. To our knowledge, WiMTR is the first multi-person mesh regression method via WiFi signals, which can serve as a reference and benchmark for subsequent research tasks.

- Our proposed WiMTR achieves centimeter-level errors, which is at the same level as previous image-based and radar-based methods.

## 2  RELATED WORK

**SMPL.** Existing human mesh regression models can be classified into two types: model-free and model-based. Model-free methods (Lin et al., 2021; Kolotouros et al., 2019b; Choi et al., 2020; Saito et al., 2020; Cho et al., 2022) directly regress the vertices of the human mesh from the input. However, these methods are challenging to extend to multi-person mesh regression due to the need to predict a large number of vertices. In contrast, model-based methods only need to predict the parameters required by a parametric model, which can then map the parameters to a human body mesh.One popular parametric human body mesh model is SMPL (Loper et al., 2015), which requires two parameters: pose parameters $\beta$ and shape parameters $\theta$. The pose parameters $\beta \in \mathbb{R}^{24 \times 3}$ represent the Axis-angle rotations of 24 body joints, while the shape parameters $\theta \in \mathbb{R}^{10}$ are the top-10 PCA coefficients in SMPL's statistical shape space. The 3D representation of the root joint denotes the global orientation of the human mesh, while the 3D representations of the remaining joints represent their rotations relative to their parent in a kinematic chain. Given these two parameters, SMPL can generate the 3D joint locations $J \in \mathbb{R}^{K \times 3}$ and the human mesh $M \in \mathbb{R}^{6890 \times 3}$. In WiMTR, we use SMPL to generate multi-person human meshes. However, instead of directly predicting

the Axis-angle expressions of joints, we estimate their 6D representations to facilitate accurate and efficient mesh regression (Zhou et al., 2019).

**3D Human Mesh from Images.** Reconstructing a single person's 3D mesh from an image is a fundamental task in camera-based human mesh regression. Zhang et al. (2020) represent the 3D human mesh via a partial UV map to handle obstructed body parts. METRO (Lin et al., 2021) directly regresses mesh vertex coordinates using the Transformer model via hundreds of vertex queries. HybrIK (Li et al., 2021) proposes a hybrid inverse kinematics solution that connects 3D pose estimation and body mesh regression. Regressing the human mesh of multiple people from monocular images is commonly accomplished using top-down methods (Li et al., 2022; Kolotouros et al., 2019a; Kanazawa et al., 2018; Kocabas et al., 2020; Cha et al., 2022). These methods transform the problem into single-person mesh regression using human detectors. However, those approaches overly rely on the accuracy of the human detectors. In contrast, ROMP (Sun et al., 2021) implements a single-stage multi-person mesh regression method based on center heatmap and 3DCrowdNet (Choi et al., 2022) uses a joint-based regressor to filter the corresponding features of the target person and alleviate the impact of crowding.

**3D Human Mesh from Radars.** The pioneering work RF-Capture (Adib et al., 2015) demonstrated the possibility of capturing human figures through walls using radio frequency (RF) signals, which has since attracted attention in the field of human pose and shape estimation. Subsequent works, RF-Pose (Zhao et al., 2018) and RF-Avatar (Zhao et al., 2019), have constructed human pose and mesh using RF signals at a more fine-grained level. These works have shown that RF signals contain enough information not only to estimate the shape of the human body, but also to overcome challenges in traditional camera-based human perception, such as occlusion, poor lighting, clothing and privacy issues. However, since these works used self-made centimeter wave radar equipment, some researchers have explored the use of commercial portable equipment for human body mesh estimation. For example, mmMesh (Xue et al., 2021) estimates the human body mesh of a single person using TI AWR1843 and $M^4$esh (Xue et al., 2022) extends this approach to multi-person mesh estimation.

**3D Human Pose and Shape Estimation from WiFi.** Similar to RF signals, WiFi signals can also obtain human body pose information in blocked or dark environments and the widespread availability of cheap hardware devices makes WiFi-based approaches attractive for human pose estimation. Single-person 2D human pose estimation based on WiFi (Wang et al., 2019a; Zhou et al., 2023) has made significant progress and some works have leveraged WiFi signals for single-person 3D pose estimation (Ren et al., 2021; 2022; Jiang et al., 2020). However, current approaches are limited to single-person scenarios due to the lack of multi-person detection algorithms. Person-in-WiFi (Wang et al., 2019b) presented the first multi-person 2D human pose estimation method based on part affinity fields (PAFs) of OpenPose (Cao et al., 2017) to group keypoints. However, due to shortcomings in the 3D encoding of PAFs, it is difficult to extend this method to the 3D multi-person domain. Although WiFi has made some achievements in human pose estimation, research on human mesh regression using WiFi is still relatively lacking. Wi-Mesh (Wang et al., 2022) proposes a method based on WiFi vision to reconstruct the human body mesh. This method uses 2D angle of arrival (AoA) to enable visualization of the perceived human body information to a certain extent using WiFi signals. Nevertheless, this method is still limited to single-person mesh regression.

## 3 METHODOLOGY

### 3.1 PREPROCESSING AND TOKENIZATION OF CSI SIGNALS

**Preprocessing.** CSI samples obtained in WiFi systems are often affected by noise and other interferences, including random phase drift and flip. As a result, some WiFi-based solutions (Wang et al., 2019b) only rely on the amplitude information and disregard the phase of the CSI signals. However, such a crude solution would clearly compromise the integrity of the information. Therefore, we use the method of unwinding and linear transformation to denoise the phase signal following PhaseFi (Wang et al., 2015). For specific details of the process, please refer to Appendix B.

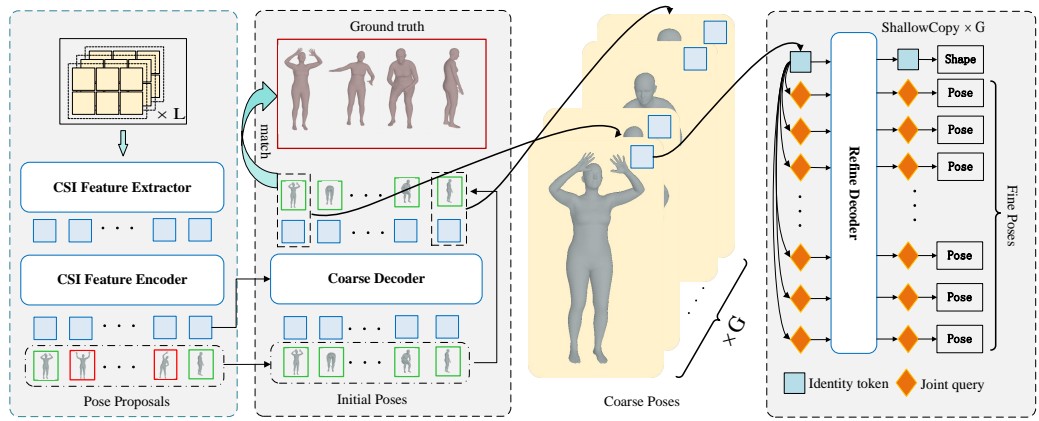

Figure 2: **The overall architecture of WiMTR.** The CSI feature extractor and CSI feature encoder are used to obtain CSI feature token sequences, which carry global information. With $N$ human queries, the coarse decoder can predict $N$ sets of mesh parameters simultaneously. The human token that matches the ground truth and $N$ joint queries are then fed into the refine decoder. The human token generates shape parameters through an MLP module, while the joint queries regress fine pose parameters through $K$ MLP modules.

The raw CSI signals $Z \in \mathbb{C}^{3\times3\times20\times30}$ are complex signals. $A \in \mathbb{R}^{3\times3\times20\times30}$ and $\Psi \in \mathbb{R}^{3\times3\times20\times30}$ are amplitude and phase of $Z$, respectively. Then we concatenate the amplitudes $A$ and denoised phases $\Psi_{denoised}$ into a token $\hat{Z} \in \mathbb{R}^{3\times3\times20\times60}$.

**Tokenization.** The raw CSI signals are transformed into an input signal with a size of $3 \times 3 \times 20 \times 60$ after the above operation processing. A question worth considering is how to input this data into the network. Previous research on human sensing with WiFi(Wang et al., 2019b) has indicated that the Convolutional Neural Network (CNN) is capable of extracting spatial features from the last two dimensions, which correspond to the $3 \times 3$ transmitting sensor pairs. Others (Geng et al., 2022) believe that each element in CSI represents a unique 1D summary of the entire scene and they flatten the signal before inputting it into the network. We propose a new way of understanding the CSI signals, where the first two dimensions ($3 \times 3$) represent spatial positions, the third dimension (20) represents temporal positions and the last dimension (60) represents features at a certain moment in a spatial position. An intuitive and visual analogy is that the first three dimensions of a CSI signal are equivalent to the width and height of the image, while the last dimension is equivalent to the channel of the image. Therefore, we flatten the first three dimensions of the signal to obtain a token sequence of size $180 \times 60$.

## 3.2 OVERVIEW

As shown in Figure 2, our model is generally divided into four parts: CSI feature extractor, CSI feature encoder, coarse decoder and refine decoder.

**CSI Feature Extractor.** We adopt a two-stage approach to extract the features of the flattened CSI signal $\tilde{Z} \in \mathbb{R}^{180\times60}$. Firstly, we extract features $F_0, F_1 \in \mathbb{R}^{180\times256}$ of amplitude $A$ and phase $\Psi$ through two independent MLPs. Subsequently, we concatenate those feature vectors and input the concatenated token $F_{total} \in \mathbb{R}^{180\times(2\times256)}$ into a feature mixer to obtain a mixed feature sequences $F_{mixed} \in \mathbb{R}^{180\times256}$, which will be input into the CSI Feature Encoder.

**CSI Feature Encoder.** The encoder consists of multiple layers, each containing a Multi-head Attention (MHA) module and a feed-forward network (FFN). In our approach, we stack six layers of encoder layers to obtain deeply encoded CSI features $F$.

In the final layer of the encoder, we introduce a module similar to the region proposal network (RPN) that provides $L$ pose proposals $\{\beta_i^{Init}\}_{i=1}^L \in \mathbb{R}^{L\times(K\times D)}$, where $L$ is the length of the sequence, $K$ is the number of joints and $D$ is the dimension of joint coordinates. At the same time, there is also a

classification branch that predicts a score for each token. These proposed poses are used as initial poses in the coarse decoder after being sampled by the score.

**Coarse Decoder.** We aim to reason a set of pose and shape parameters from the global CSI information using the coarse decoder(*i.e.*, encoded CSI feature memory $F$). The coarse decoder is also stacked by Transformer layers. Given N randomly initialized queries, the coarse decoder can output N sets of parameters, including pose parameters $\{\beta_i^C\}_{i=1}^N \in \mathbb{R}^{N \times (K \times D)}$ and shape parameters $\{\theta_i\}_{i=1}^N \in \mathbb{R}^{N \times 10}$. The physical meaning of each query is a person to be predicted.

Firstly, the human queries will be fed to a multi-head self-attention module for internal interaction. Subsequently, each human query will extract corresponding human features from encoded CSI features $F$ via a multi-head cross-attention module. Then each human-feature token will be fed into several regression heads. The classification head, a fully-connected layer (FC), will predict the confidence score of each human token. A pose regression head will estimate a set of relative offsets using a multi-layer perceptron (MLP).

Consistent with (Shi et al., 2022), we adopted a cumulative approach to predict the pose parameters. That is to say, the predicted poses of each layer are the sum of the offset predicted by this layer and the predicted poses of its previous layer. Formally, given a pose parameter $\beta_{d-1}$ predicted by the $d^{th}$ layer, the pose parameter $\beta_d$ can be represented as:

$$\beta_d = \beta_{d-1} + \Delta\beta_d \tag{1}$$

where $\Delta\beta_d$ are the offsets predict by the $d^{th}$ layer. The initial pose parameter $P_0$ are sampled from the pose proposals $\{\beta_i^{Init}\}_{i=1}^L$.

**Refine Decoder.** Inspired by some pose estimation methods such as PETR (Shi et al., 2022) and ED Pose (Yang et al., 2023), We decouple multi-person pose estimation into two stages: human-level detection and joint-level detection. The coarse decoder is responsible for extracting human-level features from CSI global context to predict a set of coarse parameters. While the refine decoder extracts joint-level features from global information to further refine the prediction of the coarse decoder. However, unlike PETR using the deformable multi-head attention module to build its joints decoder, we use a standard multi-head attention module to build refine decoder. This results in our model being unable to distinguish the differences between different people at this stage. To this end, we have introduced identity tokens $Q^{id}$.

The detailed structure of the refine decoder is shown in the Figure 3. In the refine stage, the identity token $Q^{id}$ will be concatenated with K joint queries $\{\mathcal{Q}_i^J\}_{i=1}^K$ as input $\{\mathcal{Q}_i\}_{i=1}^{K+1}$ to the refine decoder. They will first interact with each other through a multi-head self-attention mechanism. In this process, the relationship between $K$ joints will be constructed and the identity token can enable the model to obtain human identity information. Subsequently, these queries extract human-level and joint-level features through a cross attention mechanism. Finally, the query corresponding to the identity token is responsible for predicting the person's shape parameters $\theta^{fine}$, while the joint query are responsible for predicting a set of pose parameters' offset $\{\Delta\beta_i\}_{i=1}^K$. Similar to the coarse decoder, we adopt a progressive approach to obtain the final pose parameters. Refine decoder does not refine all predictions output by the coarse decoder, but instead selects tokens from the coarse decoder's prediction set that match the ground truth. During the training stage, this process is achieved through Hungarian matching between the predictions and the ground truth. In the testing stage, the selection is based on the confidence score of each token.

**Differentiation Branch.** In order to accelerate convergence speed, we don't use randomly initialized joint queries during the refine stage. We use three linear layers with residual structures to generate joint queries from the identity token.We initialize the weights and biases of the linear layer to a random distribution close to zero. In this way, before the training starts, the joint queries are almost identical to the identity token. As training progresses, joint queries will gradually differentiate into queries containing both identity and joint information.

### 3.3 LOSS FUNCTION

Similar to DETR (Carion et al., 2020), we approach the problem of regressing human mesh from WiFi signals as a set-prediction problem. To ensure that the ground truth mesh of each person has only one corresponding prediction, we apply a set-based Hungarian matching loss (Kuhn, 1955). We

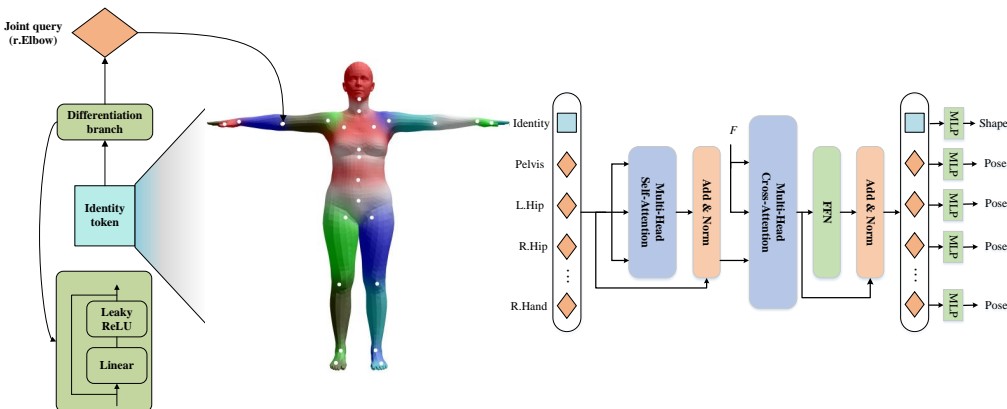

Figure 3: **The detailed structure of the differentiation branch and the refine decoder.** On the left side of the figure, we demonstrate how to differentiate specific joint queries from identity tokens using the right elbow as an example. The details of the refine decoder are shown on the right side. The identity token and joint queries are fed into the refine decoder together to refine the joints of the target person.

add constraints to the output of the Region Proposal Network (RPN) after the CSI feature encoder, the output of each layer of coarse and refine decoder. The total loss function comprises five parts: classification loss ($L_{cls}$), keypoint loss ($L_{kpt}$), pose parameter loss ($L_{pose}$), shape parameter loss ($L_{shape}$) and joint loss ($L_{joint}$). We use focal loss (Lin et al., 2017) as the classification loss and $L2$ loss as the remaining loss. The total loss function is formulated as:

$$L_{total} = \lambda_{cls}L_{cls} + \lambda_{kpt}L_{kpt} + \lambda_{pose}L_{pose} + \lambda_{shape}L_{shape} + \lambda_{joint}L_{joint} \qquad (2)$$

Where $\lambda_{cls}$, $\lambda_{kpt}$, $\lambda_{pose}$, $\lambda_{shape}$ and $\lambda_{joint}$ are the loss weights for $L_{cls}$, $L_{kpt}$, $L_{pose}$, $L_{shape}$ and $L_{joint}$, respectively. To accelerate convergence, we add an auxiliary keypoint detection branch for training. This branch directly returns the 3D coordinates of human body keypoints. $L_{joint}$ is the $L2$ loss between the joints output by SMPL and the ground truth labels. $L_{kpt}$ is the loss function used to constrain the keypoint detection branch. To avoid unstable training, we calculate the loss of pose in the $3 \times 3$ rotation matrix format. The specific formulas for each loss function are listed in Appendix C.

## 4 EXPERIMENTS

### 4.1 DATASET

We use 4 Lenovo ThinkPad X201 laptops equipped with Intel 5300 network cards to collect CSI data and a Microsoft Azure Kincet camera to collect RGB images. The equipments and scenario for collecting data are shown in Figure 1. More details about the dataset can be found in Appendix A.

### 4.2 TRAINING DETAILS

We train WiMTR on 8 NVIDIA GeForce RTX 3090 GPUs with AdamW (Loshchilov & Hutter, 2017) optimizer with base learning rate of $6 \times 10^{-5}$, momentum of 0.9 and weight decay of $1 \times 10^{-4}$. We trained the model for a total of 100 epochs and the learning rate will decay to 0.1 of the initial learning rate at the $80^{th}$ epoch. The loss weights are set to $\lambda_{cls} = 20$, $\lambda_{kpt} = 100$, $\lambda_{pose} = 40$, $\lambda_{shape} = 3$ and $\lambda_{kpt} = 100$.

## 4.3 EVALUATION METRICS

In order to compare with other mesh regression methods, we follow settings of ROMP (Sun et al., 2021) during testing. We adopt per-vertex error(PVE) to measure the 3D surface error of the human mesh. In addition, we also employ mean per joint position error(MPJPE) and Procrustes-aligned MPJPE(PMPJPE) to measure joint errors. When calculating MPJPE and PAMPJPE, we use the 14 joints annotated in the LSP (Johnson & Everingham, 2010; 2011) dataset.

## 4.4 RESULTS

To quantitatively evaluate the performance of our model, we present several experimental results in Table 1. As the first work to reconstruct multi-person human meshes from WiFi signals, there is no prior work to compare against directly. Nonetheless, we also include results from mesh regression using other modalities for reference. It should be noted that these results are not directly comparable to ours, as the datasets used in these methods are different. Nevertheless, these results demonstrate that our approach using WiFi signals for mesh regression is highly competitive and holds great promise for future research in this field.

Table 1: **Comparison with existing work.** Our work achieved centimeter-level error, which is a competitive result compared to previous work. WiMTR is the only multi-person mesh regression method based on WiFi. [†] represents that this method supports multi-person scenarios.

| Work | Dataset | Modality | MPJPE | PAMPJPE | PVE |
|---|---|---|---|---|---|
| ROMP[†] (Sun et al., 2021) | 3DPW | RGB | 76.7 | 47.3 | 93.4 |
| CLIFF[†] (Li et al., 2022) | 3DPW | RGB | 69.0 | 43.0 | 81.2 |
| METRO (Lin et al., 2021) | 3DPW | RGB | 77.1 | 47.9 | 88.2 |
| 3DCrowdNet[†] (Choi et al., 2022) | 3DPW-Crowd | RGB | 85.8 | 55.8 | 108.5 |
| FastMETRO (Cho et al., 2022) | 3DPW | RGB | 73.5 | 44.6 | 84.1 |
| mmMesh (Xue et al., 2021) | mmMesh Dataset | mmWave | 21.8 | - | 24.7 |
| M$^4$esh[†] (Xue et al., 2022) | M$^4$esh Dataset | mmWave | 40.1 | 18.4 | 43.1 |
| Wi-Mesh (Wang et al., 2022) | Wi-Mesh Dataset | WiFi | 24.0 | - | 28.1 |
| WiMTR[†](ours) | our dataset | WiFi | 71.4 | 29.7 | 57.3 |

We present the performance of our model under different population conditions in Figure 2. Specifically, we evaluate the error of WiMTR in scenes containing 1, 2, 3 and 4 people. The results show that our model can achieve centimeter-level error regardless of the number of people in the scene, demonstrating its ability to handle multi-person scenarios. This is particularly important in real-world applications, where it is common to have multiple people present in a scene.

Table 2: **Errors in different numbers of person.**

| | MPJPE | PAMPJPE | PVE |
|---|---|---|---|
| one person | 53.0 | 37.2 | 62.7 |
| two person | 67.3 | 24.0 | 82.6 |
| three person | 82.1 | 28.8 | 100.9 |
| four person | 97.2 | 61.1 | 119.3 |

## 4.5 ABLATION STUDY

To evaluate the effectiveness of the modules designed in our model, we conducted several ablation experiments. Due to time constraints, we trained the model for only 30 epochs during the ablation experiments, with all other settings remaining unchanged. The results of the ablation study are presented in Table 3.

**Refine Decoder.** The refine decoder aims to adjust the output of the coarse decoder and generate a set of more refined parameters. To demonstrate the effectiveness of the refine decoder, we conducted an experiment where we removed the refine decoder and used only the rough pose and shape parameters output by the coarse encoder. The results showed that after removing the refine decoder, the MPJPE increased by 1.7mm, the PAMPJPE increased by 1.4mm and the PVE increased by 1.7mm, which is consistent with our expectations for the refine decoder.

**Joint Differentiation.** In the refine decoder, we introduced a novel joint query generation method called differentiation, which generates joint queries based on the identity token using a residual linear module. At the start of training, all joint queries are identical to the identity token. However, as training progresses, joint queries gradually differentiate into queries with different joint information, allowing us to obtain a set of customized queries with the identity information of the owner. We compared the results obtained from queries generated by the differentiation method with those obtained from randomly initialized joint queries and found that using learnable joint queries can result in an increase of 0.5mm in MPJPE, 0.5mm in PAMPJPE and 0.5mm in PVE. These results highlight the importance of our differentiation method and demonstrate its effectiveness.

**Phase Sanitization.** To quantitatively analyze the effect of phase processing, we conducted three experiments under equivalent conditions: one without phase input, one with raw phase input and one with denoised phase input. The experiments demonstrate that the absence of phase information in the input leads to increases of 1.6mm in MPJPE, 1.3mm in PMPJPE and 2.0mm in PVE. Using raw phase information results in increases of 1.0mm in MPJPE, 0.9mm in PMPJPE and 1.0mm in PVE. These results illustrate the significance of utilizing denoised phase information.

Table 3: **Ablation Study.** To validate the effectiveness of various designs presented in our paper, we conducted several sets of experiments using control variates.

| Refine Decoder | Joint Differentiation | Phase | MPJPE | PMPJPE | PVE |
|:---:|:---:|:---:|:---:|:---:|:---:|
| ✗ | - | denoised | 78.6 | 32.4 | 96.0 |
| ✓ | ✗ | denoised | 77.4 | 31.5 | 94.8 |
| ✓ | ✓ | ✗ | 78.5 | 32.3 | 96.3 |
| ✓ | ✓ | raw | 77.9 | 31.9 | 95.3 |
| ✓ | ✓ | denoised | 76.9 | 31.0 | 94.3 |

### 4.6 QUALITATIVE ANALYSIS

We demonstrate the predictions of our model in various population and scenarios, as shown in Figure 4. The visualization results indicate that WiMTR can efficiently regress both pose and shape parameters from CSI signals. This demonstrates that despite being a sparse signal with minimal data, CSI still contains the necessary information for accurately reconstructing the human mesh. This achievement holds heuristic significance for future research in human mesh regression from WiFi signals.

Although our labels are generated using ROMP (Sun et al., 2021), WiMTR learns the true mapping from CSI signals to the human mesh, rather than the distribution of ROMP's output domain. Figure 5 demonstrates an example where ROMP makes a highly incorrect prediction about the arm of the person in the middle. In contrast, WiMTR's predicted pose, while smaller than the ground-truth, still accurately captures the information that the arm is lifted.

## 5 CONCLUSIONS

In this paper, we demonstrate the feasibility of reconstructing multi-person 3D human mesh from WiFi signals. We introduce the first multi-person mesh dataset based on CSI signals, which includes 171,183 frames of data. Our method achieves the same level of error as previous methods based on other signals, with MPJPE of 71.4mm, PAMPJPE of 29.7mm and PVE of 57.3mm on this dataset. These results can serve as a baseline for future research on this dataset.

The dataset proposed in this paper is the first multi-person mesh dataset based on WiFi, providing a convenient resource for future related work. However, it is important to note that this dataset has

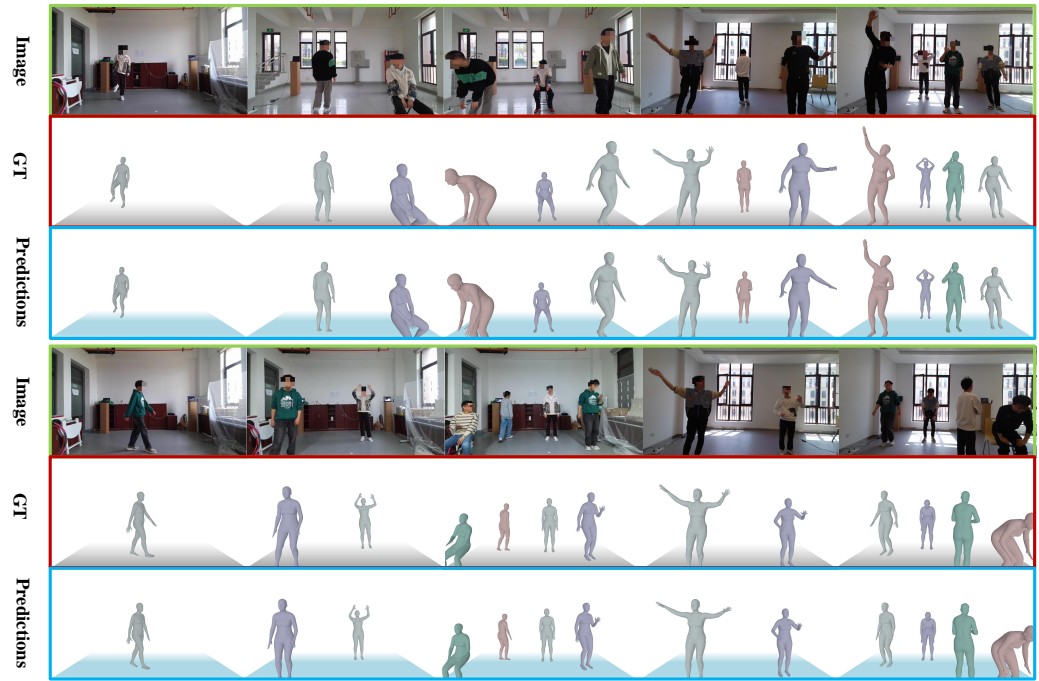

Figure 4: Some 3D Huamn Mesh Regression Results. We visualize the predicted meshes in three different scenarios, covering the situation of one to four people. The first row is the image captured by the camera, the second row is the ground-truth of the meshes and the third row is the meshes predicted by WiMTR.

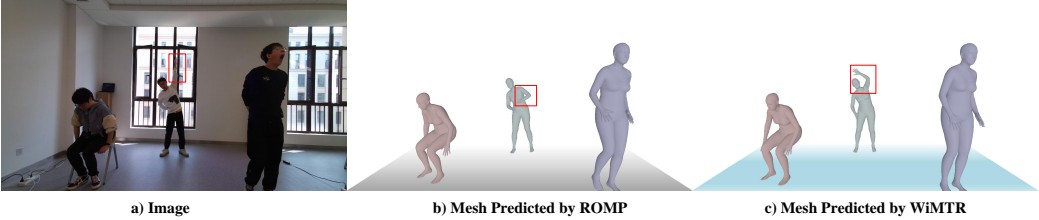

Figure 5: In this example, ROMP confuses the target's arm with the window frame and mistakenly identifies the pattern on the clothes as the arm. In contrast, WiMTR accurately captures the lifting of the arm.

some limitations. Due to the lack of relevant mesh collection equipment, we generated our mesh labels using state-of-the-art camera-based mesh regression methods (Sun et al., 2021). Although we manually screened out the vast majority of samples with inaccurate predictions, there may still be a small number of bad labels in this dataset. Additionally, the limitations of the camera-based method result in a lack of occluded training data, which may limit the model's ability to generalize to occluded scenarios.

Despite the limitations of the dataset, we believe that our work is still significant. The proposed dataset and WiMTR model represent a crucial step forward in the WiFi-based approach. We acknowledge that there is always room for improvement and further exploration and we hope that our work can inspire and motivate future research efforts.

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

## A  DETAILS OF THE DATASET

We introduce the first WiFi-based multi-person mesh regression dataset, which was collected using 4 Lenovo ThinkPad X201 laptops with Intel 5300 network cards and a Microsoft Azure Kinect camera. Seven volunteers performed various actions in 3 scenarios, namely an office room, a classroom and a corridor (as shown in Figure 6). The data collection process lasted for one week, during which we collected a total of 415,220 frames of raw data, consisting of both CSI signals and RGB videos. The labels of the dataset were generated from the RGB videos using a camera-based method (Sun et al., 2021), but there were many inaccurate predictions that could affect model training. To address this, we recruited nearly 20 students to manually screen the dataset frame by frame, which took us almost a week to complete. After cleaning, a total of 171,183 valid data frames were obtained. The statistical information of the data is shown in the Table 4.

Table 4: Dataset statistics.

|          | 1-person | 2-person | 3-person | 4-person | all     |
|----------|----------|----------|----------|----------|---------|
| training | 22,061   | 80,746   | 37,558   | 11,473   | 151,838 |
| test     | 2,827    | 9,595    | 5,623    | 1,300    | 19,345  |

One of these four laptops serves as the signal transmitter and the other three serve as signal receivers. The transmitter has 1 antenna responsible for broadcasting WiFi packages, which can send 300 packages per second. Three receivers use three antennas to monitor broadcast signals. Thus a Linux CSI tool (Halperin et al., 2011) collects CSI tensors $\in \mathbb{C}^{3 \times 3 \times 30 \times 300}$ per second. At the same time, the camera will collect RGB videos from the corresponding time period to generate annotations. The video is 15 fps. Therefore, each frame of the image is aligned with a CSI data of size $3 \times 3 \times 30 \times 20$. We transpose the last two dimensions of the CSI signal before inputting it into the model and process their amplitudes $A \in \mathbb{R}^{3 \times 3 \times 20 \times 30}$ and phases $\psi \in \mathbb{R}^{3 \times 3 \times 20 \times 30}$ separately.

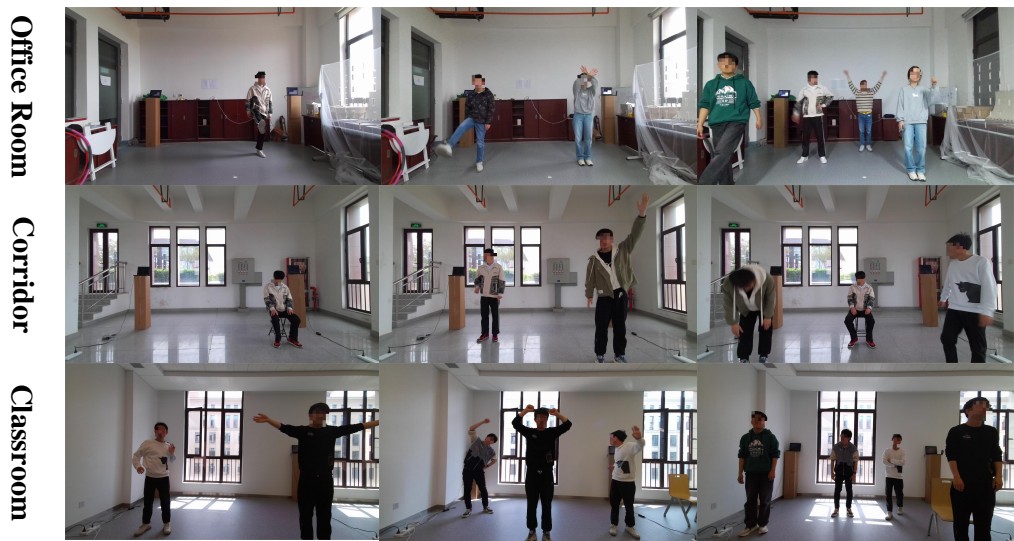

Figure 6: **Schematic diagram of the data collection scene.** To ensure the diversity of the data, we conducted data collection in three scenes: office room, corridor and classroom. In each scene, we collected data with 1 to 4 people present, respectively.

## B DETAILS OF PHASE SANITIZATION

CSI signals' phase information is often affected by noise and other factors, such as drift and flipping. Therefore, some CSI-based methodss (Wang et al., 2019b) do not use phase information as input, which inevitably compromises the integrity of the information. To address this issue, we adopted a phase sanitization method proposed in PhaseFi (Wang et al., 2015). Although this is not our contribution, we still present some analysis and visualization results in this section to demonstrate the importance of this operation. It is mentioned in this paper that the measured CSI signal can be represented by the following formula:

$$\tilde{\theta}_i = \theta_i + \frac{2\pi i \Delta t}{N_{ce}} + \beta + Z \tag{3}$$

where $i$ represents the subcarrier index; $\hat{\theta}_i$ and $\theta_i$ represent the measured CSI signals and the real CSI signals, respectively; $\Delta t$ is the time lag; $N_{ce}$ is a constant, e.g., $N_{ce} = 64/128$ for a bandwidth of 20/40MHz; $\beta$ represents the random phase offset due to the device's own defects and $Z$ is random noise during the measurement process. The phase signals are first unwrapped to prevent it from flipping at the boundaries. For the monitors, the two noise terms in the middle of the equation3 remain constant during transmission. Linear phase denoising introduces two variables that are only dependent on the real phase and are consistent across each subcarrier during transmission:

$$a = \frac{\tilde{\theta}_n - \tilde{\theta}_1}{n - 1} \tag{4}$$

$$b = \frac{1}{n} \sum_{i=1}^{n} \tilde{\theta}_i \tag{5}$$

The phase after denoising can be represented by the following formula:

$$\hat{\theta}_i = \tilde{\theta}_i - a k_i - b = \tilde{\theta}_i - \frac{\tilde{\theta}n - \tilde{\theta}_1}{n - 1} i - \frac{\sum_{i=1}^{n} \tilde{\theta}_i}{n} \tag{6}$$

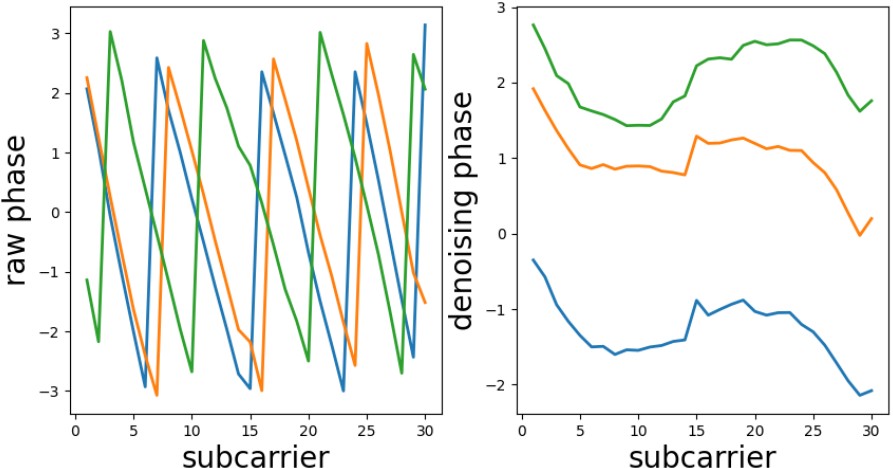

Figure 7: **Waveform of the phase signals.** We selected the first group of subcarrier signals of the first antenna in the three monitors to show the effect of the phase sanitization. The left shows the waveform of the raw phase signals, while the right shows the waveform of the phase signals after denoising.

Figure 7 visualizes the raw phase and the denoised phase. The results indicate that after processing, the phase information becomes continuous and easy to distinguish. Section 4.5 provides detailed quantitative experiments regarding phase.

## C   FORMULA FOR THE LOSS FUNCTIONS

To address the issue of imbalanced positive and negative samples, we employed the Focal Loss (Lin et al., 2017) as the classification loss. It can be expressed by the following formula:

$$L_{cls} = \text{FL}(p_t) = -\alpha_t (1 - p_t)^\gamma \log(p_t)$$

Here, if $y = 1$, $p_t = \hat{p}$; otherwise, $p_t = 1 - \hat{p}$. $\alpha_t$ is used to adjust the importance of positive and negative samples. Here, if $y = 1$, $\alpha_t = \alpha$; otherwise, $\alpha_t = 1 - \alpha$. The value of alpha has been set to 0.25. The $\gamma$ is an adjustable factor and we set it to 2.0.

The remaining regression losses($L_{kpt}, L_{pose}, L_{shape}, L_{joint}$) are all L2 Loss.

## D   ANALYZING THE CONTRIBUTION OF THE DIFFERENTIATION BRANCH

As mentioned earlier, if all queries in the refine decoder, including human queries and joint queries, are learnable queries, then the output offset of the decoder is the same when refining different human poses. This is because the input queries, keys and values to the refine decoder are the same for all individual entities that require refinement and do not include any specific information about the person being refined. One solution is to use the tokens output by the coarse decoder as the human queries and learnable queries as the joint queries. However, in this way, the joint queries can only interact with the human query in the self-attention module to obtain the individual identity information. To address this, we adopt a differentiation approach to generate the joint queries from the human query. This branch is an MLP with a residual structure. This design ensures that all joint queries are the same as the human query at the beginning of training. As training progresses, the joint queries gradually differentiate into ones that carry joint-specific characteristics. Figure 8 shows the PCA analysis of the human query and joint queries at the 0-th, 30-th, 60-th and 100-th epochs. The result shows how the joint queries gradually differentiate from the human query over the training process. Figure 9 illustrates the distribution of joint queries generated from four different identity tokens. It is evident that these queries are divided into four distinct clusters. These results demonstrate that the joint

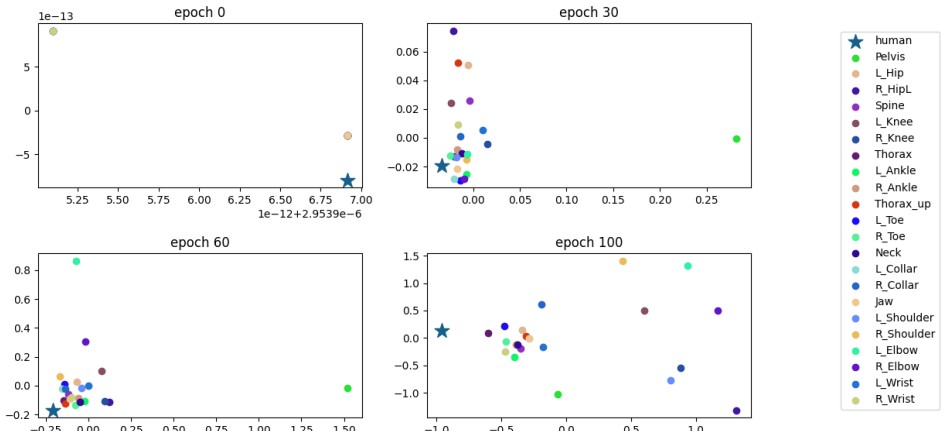

Figure 8: **PCA analysis of queries at different epochs.**

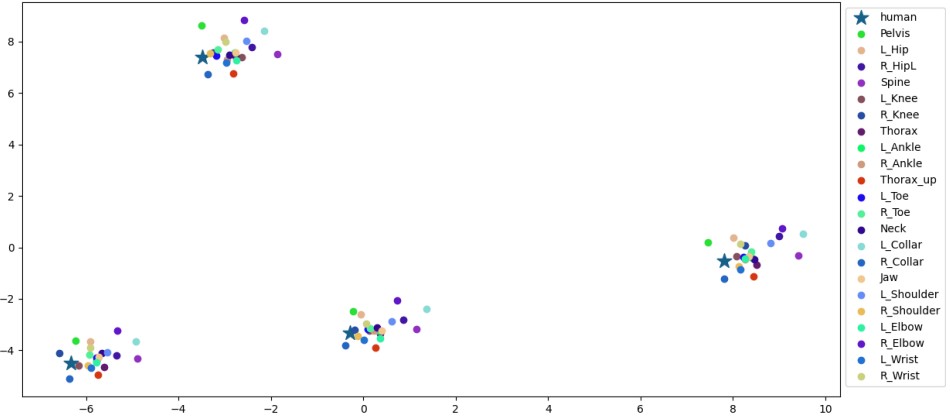

Figure 9: **PCA analysis of queries from different individuals.** We use the differentiation branch at the 100th epoch to generate joint queries for four different individuals. It can be clearly seen that these queries are divided into four different clusters.

queries generated through the differentiation branch have two advantages over randomly initialized joint queries: 1) they possess both human and joint characteristics, enabling more precise refinement of joint parameters for specific individuals; 2) they avoid starting training from randomly initialized parameters, resulting in faster convergence.

## E  MESH REGRESSION UNDER OCCLUSION AND DARK CONDITIONS.

Camera-based mesh regression methods are vulnerable to occlusion and lighting conditions. In contrast, WiFi-based sensors are not limited by these factors, which is one of their advantages over RGB sensors. To verify this, we compared the predictions of the two methods under scenarios with occlusion and dark conditions, respectively. Figure 10 shows the experimental results under occlusion conditions. During the data collection process, one person was obstructed by a white screen, while the other person was visible to the camera. As expected, the camera-based method fails to predict the person who was not visible. In contrast, our WiMTR successfully predicts the person who was not visible. Figure 11 shows the experimental results under dark conditions. In very low-light conditions,

the camera-based method is able to detect the presence of a person but can not accurately predict their specific pose or global orientation. In contrast, lighting conditions do not affect WiFi signals and WiMTR is able to regress the mesh of the person without any difficulty.

| Image | RGB Prediction | WiFi Prediction | Lateral View |
|---|---|---|---|

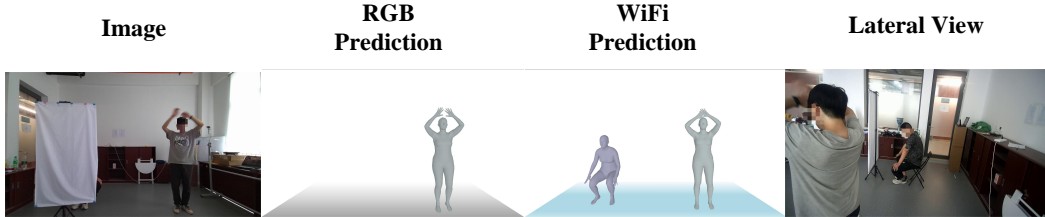

Figure 10: **Experimental results under occlusion conditions.** We present the image captured by the Microsoft Azure Kinect camera, along with the predictions of the camera-based method and WiMTR. To show the person behind the white screen, an additional camera was placed on the side and its image is displayed in the final column.

| Image | RGB Prediction | WiFi Prediction |
|---|---|---|

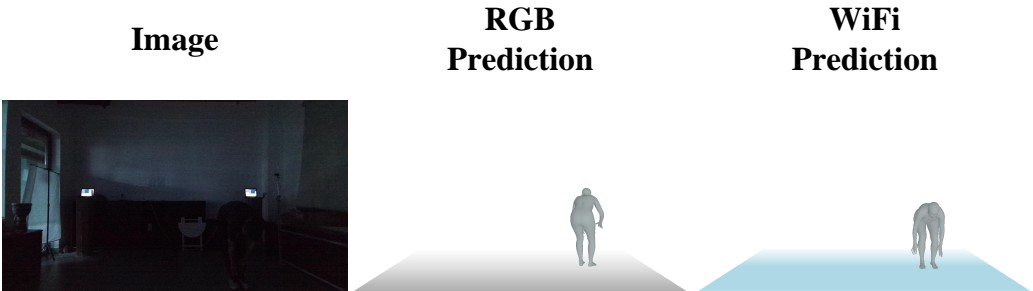

Figure 11: **Experimental results under dark conditions.**

## F  MORE VISUALIZATION

In order to demonstrate the effectiveness of our model more comprehensively, we have provided more visualization results in Figure 12.

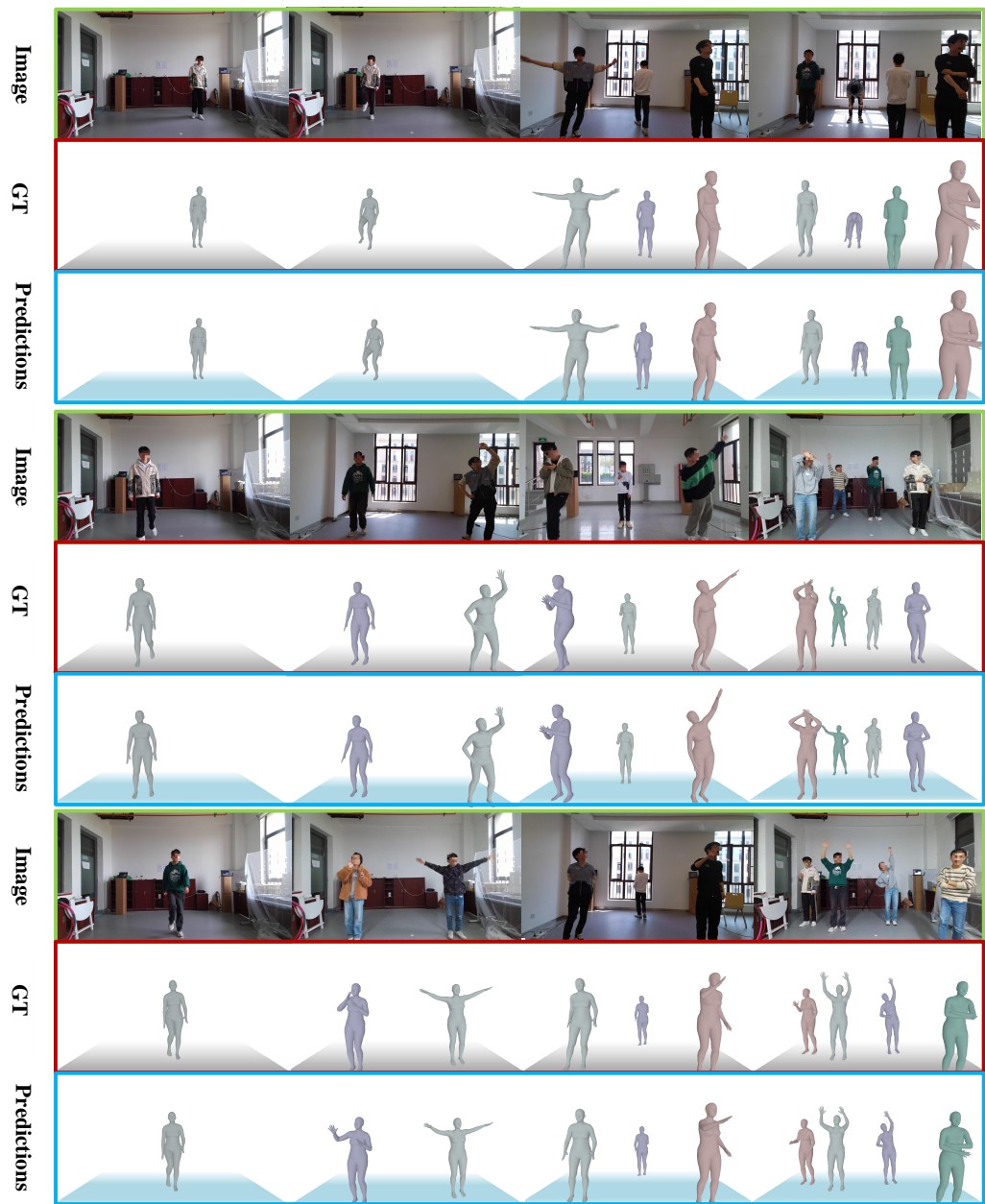

Figure 12: **More Visualization Results.**

