# OpenReview forum: "From Sparse to Dense: Learning to Construct 3D Human Meshes from WiFi"
_ICLR.cc/2024/Conference — Submitted to ICLR 2024_

### Official Review · Reviewer_d8sA · 2023-10-30

**Soundness:** 1 poor
**Presentation:** 2 fair
**Contribution:** 1 poor
**Rating:** 1
**Confidence:** 4

**Summary:**

The authors propose a WIFI pose regression method based on a transformer architecture. In contrast to the state of the art, they are able to track multiple persons simultaneously.

**Strengths:**

- Reconstructing meshes using WiFi signals is new and worth investigating
- The datasets will be published.

**Weaknesses:**

There are several weaknesses in the paper in its current form:
1. For the greater part of the ICLR audience WIFI signals are not a daily business. The paper lacks an intuitive explanation of what information is contained in CSI and what’s not. The paper talks a lot about tensor dimensions which are not clearly justified and explained. The inputs and outputs of each component of the models do not become clear.
2. There are not enough details about the hardware topology and the used bandwidth. There is only a statement “…a bandwidth of 20/40MHz…” in Section B. However, you did not describe, which bandwidth is used. Also, a detailed discussion, why a radio system with only 20/40MHz can achieve such high accuracies is missing. Such high localization results can even not be achieved by active, WIFI based localization systems, also working with OFDM signals. Why should your passive approach be even better than those?
3. We also miss a comparison of other algorithms on your dataset, as well as your method evaluated on publicly available datasets. The results shown in Table 1 cannot be used to compare your results to the state of the art, this is unfair. The accuracy is highly correlated to the quality of the dataset. We acknowledge difficulty in comparing the datasets, but WiTR could still be compared with one of the single-person methods on the dataset that the paper proposes.
4. The paper aims at providing novelty in mesh reconstruction. At the algorithmic level however, there is not much contribution.
5. It is unclear how such a system works out of the lab. I am less concerned about real-time processing but the CSI data itself. How well does it generalize to unseen trajectories, movements etc. It is totally unclear how the training and test data are organized.

**Some minor comments:**

- “…which typically consists of only about 10,000 elements if…” - This depends highly on the used radio system, employed bandwidth and topology. There is no “typical” size of a CSI.
- “...meshes (with a size of N × 6890 × 3) representing…” – What is “N”?
- “…which is at the same level as previous image-based and radar-based methods” - I cannot find any comparison to such methods in your work. The accuracy highly depends on the environment and scenario. A direct comparison of accuracies is not useful.
- “…random phase drift and flip.” - You may explain what a phase flip is.
- “…linear transformation to denoise the phase signal following…” - How did you deal with the phase shift and drift among the four network cards?
- “…CSI signals Z ∈ C3×3×20×30 are complex signals.”  - What do the individual dimensions stand for?
- “We propose a new way of understanding the CSI signals, where…” - Your understanding of the CSI is not in line with the communication theoretical definition.
- “…collects CSI tensors ∈ C3×3×30×300 per second.” - What is the last dimension and why do you only use 20 samples of the 300 and which one of them? You must explain the recorded data detailed and reason why you can just crop the dimension.
- How big is the label noise resulting from the employed camera-based mesh regression?
- What is G in Fig. 2? It seems to be the number of coarse poses but how is it defined?
- While Geng et al. 2022 in general being different from the proposed work (i.e., not working on mesh models) it would be beneficial to also differentiate from it in the paper.
- Intro: “We present a[-n-] fully end-to-end
- Sec. 2: “body mesh.[ ]One”
- Sec. 2: what is a top-down method?
- Sec. 3.: “sensing with WiFi[ ](Wang)”
- Sec. 3: “coarse decoder[ ](i.e.,…”
- Sec. 3: “identity token.[ ]We”
- Sec 3.: [w]here after Equation 2
- Table 1: it is not immediately clear what the unit is for the mentioned results


**Appendix A**

How did you separate Training and Test data? Are they different recordings on different days or just random splits? A random split is not a valid evaluation. You have to share more details about the generation of the datasets to emphasize the generalization abilities and high localization results.

**Questions:**

See above.

---

### Official Review · Reviewer_UpgU · 2023-10-31

**Soundness:** 2 fair
**Presentation:** 4 excellent
**Contribution:** 2 fair
**Rating:** 5
**Confidence:** 5

**Summary:**

The paper proposes a transformer-based network for WiFi-based mesh reconstruction. The paper proposes to estimate the mesh of human bodies by transferring supervision from camera to the WiFi CSI data. Though the CSI data is quite sparse, the results show that the proposed approach achieves good results on a self-collected dataset.

**Strengths:**

1. The paper is well written and organized.
2. The task is quite new and the gap is well illustrated in WiFi sensing.
3. The proposed method achieves acceptable results on a self-collected dataset.

**Weaknesses:**

1. The technical novelty is not sufficient. The proposed method for WiFi-based mesh construction relying on camera-based supervision has been well explored [1][2][3].
2. The proposed method seems to be overfitting to one environment.  It is not realistic to deploy multiple pairs of WiFi for this specific task.
3. The experiments are conducted in a self-collected manner, which is good, but for AI conferences, public datasets including more environments should be utilized to validate the effectiveness of this work, e.g., the MM-Fi [4].

[1] Geng, Jiaqi, Dong Huang, and Fernando De la Torre. "DensePose From WiFi." arXiv preprint arXiv:2301.00250 (2022).
[2] Wang, Yichao, Yili Ren, Yingying Chen, and Jie Yang. "Wi-Mesh: A WiFi Vision-based Approach for 3D Human Mesh Construction." In Proceedings of the 20th ACM Conference on Embedded Networked Sensor Systems, pp. 362-376. 2022.
[3] Wang, Yichao, Yili Ren, Yingying Chen, and Jie Yang. "A wifi vision-based 3D human mesh reconstruction." In Proceedings of the 28th Annual International Conference on Mobile Computing and Networking, pp. 814-816. 2022.
[4] Yang, Jianfei, He Huang, Yunjiao Zhou, Xinyan Chen, Yuecong Xu, Shenghai Yuan, Han Zou, Chris Xiaoxuan Lu, and Lihua Xie. "MM-Fi: Multi-Modal Non-Intrusive 4D Human Dataset for Versatile Wireless Sensing." NeurIPS-23 Datasets and Benchmarks Track

**Questions:**

1. Could the authors further clarify the technical novelty of the proposed network? How does it contribute to the WiFi-based mesh reconstruction field?
2. Does the system work in multiple environments, i.e., cross-environment evaluation (training the model in one environtment and testing it in another one)? More experiments should be supplemented including the public dataset validation.
3. I see that the phase sanitization does not improve a lot. What are the possible reasons?
4. How are the hyper-parameters in the network decided, e.g., the layers or heads of transformers? Will deeper model achieve better performance or it has been sufficient? The authors need to clarify the reason behind the design.
5. It is recommended to use some public datasets for better demonstration.

---

### Official Review · Reviewer_XjCG · 2023-10-31

**Soundness:** 3 good
**Presentation:** 3 good
**Contribution:** 2 fair
**Rating:** 5
**Confidence:** 4

**Summary:**

This paper proposes a multi-person mesh dataset based on WiFi signals, as well as an end-to-end multi-person mesh regression method via WiFi signals, which achieves centimeter-level errors and can serve as a reference and benchmark for subsequent research tasks.

**Strengths:**

This paper demonstrates the feasibility of reconstructing multi-person 3D human mesh from WiFi signals and introduces the first multi-person mesh dataset based on CSI signals, which provides a convenient resource for future related work. Besides, it proposes a benchmark that can achieve centimeter-level errors.

**Weaknesses:**

This paper uses a camera-based method to generate labels from RGB videos. Although the paper states that nearly 20 students were recruited to manually screen the dataset frame by frame for almost a week, the quality of the dataset can still be questioned. No other methods were used for experiments on the proposed dataset except for WiMTR to evaluate the quality of the proposed dataset.

The motivation of the work is unclear. This paper does not fully emphasize the differences between this work and the single-person mesh regression method and dataset from WiFi, such as Wi-Mesh[1]. The scenes of the proposed dataset are simple. The three scenes are very similar, with a small number of people scattered. Besides, there is no interaction between people and the distances between people and sensors are also very close. The design and results of experiments do not fully reflect the strengths of multi-person.

There are some imprecise parts in this paper. For example, the paper mentions that the shape parameters are generated in the Coarse Decoder stage but does not mention how to generate them. This paper uses 𝜃 to represent shape parameters and 𝛽 to represent pose parameters, which are contrary to the conventional representations.

The authors organized related work based on the type of sensors. However, LiDAR-based MoCap is missing. For example, LiDARCap[2] is a LiDAR-based human motion estimation method.

Grammar mistakes exist. For example,‘ We present an fully end-to-end WiFi-based Mesh Regression Transformer’ in the second point of main contributions in the introducition part should be ’ We present a fully end-to-end WiFi-based Mesh Regression Transformer’.

References
[1] Yichao Wang, Yili Ren, Yingying Chen, and Jie Yang. Wi-mesh: A wifi vision-based approach for 3d human mesh construction. In Proceedings of the 20th ACM Conference on Embedded Networked Sensor Systems, pp. 362–376, 2022.
[2] Li et al., LiDARCap: Long-range Marker-less 3D Human Motion Capture with LiDAR Point Clouds, CVPR 2022

**Questions:**

This paper uses ROMP to generate labels, but experimental results in Table 1 show that ROMP performs worse than CLIFF and FastMETRO, so why use ROMP to generate labels instead of using other better methods?
Can the annotations of the dataset be improved? It is recommended to qualitatively and quantitively evaluate the quality of the annotation process.
How and why does the Refine Decoder select tokens from the N sets of parameters output by Coarse Decoder? Doesn't the N sets of parameters output by Coarse Decoder already represent N individuals to be predicted?

**Details Of Ethics Concerns:**

There are no suspicious ethical concerns.

---

### Official Review · Reviewer_gDFT · 2023-11-01

**Soundness:** 3 good
**Presentation:** 2 fair
**Contribution:** 3 good
**Rating:** 5
**Confidence:** 3

**Summary:**

In this paper, the authors focus on the problem of 3D multi-person mesh reconstruction using the Channel State Information (CSI) from WiFi signals. First, the authors create their own dataset which consists of 171,183 frames for three scenarios and supports up to 4 people. Then, the authors propose an end-to-end multi-person mesh regression transformer that utilizes a coarse-fine mechanism. The coarse decoder follows the DETR style and generates human-level pose candidates, while the refiner takes in each candidate and performs joint-level refinement. Using the proposed method, the authors can get centimeter-level errors that are comparable with other frameworks using RF signals or RGB images.

**Strengths:**

1. The authors propose the first dataset for WiFi-based multi-person 3D mesh reconstruction. At the same time, this is also the first paper that studies to use sparse CSI for 3D human mesh reconstruction. So I think this paper definitely benefits future research in this area

2. The proposed method shows promising results based on the numbers and visualizations in the paper

**Weaknesses:**

1. The novelty of the proposed multi-person mesh regression transformer is somewhat limited as it follows the same workflow as PETR [1], a prior work for multi-person pose estimation.

2. This paper is not written clearly in general and it causes some confusion while reading (see questions).


[1] Shi, Dahu, et al. "End-to-end multi-person pose estimation with transformers." Proceedings of the IEEE/CVF Conference on Computer Vision and Pattern Recognition. 2022.

**Questions:**

1. What is the 'feature mixer' in CSI Feature Extractor? What are the benefits of it?

2. In section 3.2, the authors define several parameters such as L, K, D, and N. What are their values used in the experiments?

3. In the Refine Decoder, how to get the identity tokens Q^{id}? It cannot be clearly seen in Figure 3 and there lacks related explanations in the text. Besides, there is no ablation study on identity tokens to show its effect.

4. The authors mention that the coarse decoder outputs a set of shape parameters {theta_i}. How are they used in the Refine Decoder? Do the shape parameters experience similar accumulative refinement as the pose parameters?

5. The loss function is not well defined. What is the classification loss? Is it related to the confidence score of the Coarse Decoder? Also, how is the auxiliary keypoint detection branch added to the proposed method? What is the definition of L_{kpt}?

6. The collected dataset contains three scenarios. Are there three models trained on each scenario or just one model trained on all three scenarios? If each scenario has its own model, it would be better to show the numbers for different scenarios separately.

---

### Meta-Review · Area_Chair_Cg5r · 2023-12-06

**Metareview:**

The paper presents a novel approach to 3D multi-person mesh reconstruction using WiFi Channel State Information (CSI). The authors first create a substantial dataset and then propose an end-to-end multi-person mesh regression transformer with a coarse-fine mechanism. The coarse decoder generates human-level pose candidates following the DETR style, while the refiner performs joint-level refinement. Despite the sparsity of CSI data, the proposed method achieves centimeter-level accuracy, comparable to existing frameworks using RF signals or RGB images.

Despite the authors' vast efforts, unfortunately, the reviewers were not convinced by the submission and left reasonable comments.
However, no rebuttal was summited by the authors. Thus, no chance to be accepted given conditions.

**Justification For Why Not Higher Score:**

No rebuttal was summited by the authors.

**Justification For Why Not Lower Score:**

.

---

### Decision · Program_Chairs · 2024-01-16

Reject